# A Four-Probiotic Regime to Reduce Surgical Site Infections in Multi-Trauma Patients

**DOI:** 10.3390/nu14132620

**Published:** 2022-06-24

**Authors:** Georgios Tzikos, Despoina Tsalkatidou, George Stavrou, Giannoula Thoma, Angeliki Chorti, Maria Tsilika, Antonios Michalopoulos, Theodosios Papavramidis, Evangelos J. Giamarellos-Bourboulis, Katerina Kotzampassi

**Affiliations:** 11st Propedeutic Department of Surgery, Aristotle University of Thessaloniki, AHEPA University Hospital, 54636 Thessaloniki, Greece; tzikos_giorgos@outlook.com (G.T.); despinasani@gmail.com (D.T.); chorange2404@gmail.com (A.C.); amichal@auth.gr (A.M.); papavramidis@hotmail.com (T.P.); 2Department of Surgery, Aristotle University of Thessaloniki, 54636 Thessaloniki, Greece; stavgd@gmail.com; 3Leeds Institute of Emergency General Surgery, Leeds Teaching Hospitals NHS Trust, Leeds LS97LS, UK; 4Intensive Care Unit, Aghios Pavlos General Hospital, 55134 Thessaloniki, Greece; tzenithoma@yahoo.gr; 54th Department of Internal Medicine, National and Kapodistrian University of Athens, Medical School, 12462 Athens, Greece; martsili@yahoo.gr (M.T.); egiamarel@med.uoa.gr (E.J.G.-B.)

**Keywords:** surgical site infection, probiotics, trauma, brain trauma, intensive care unit

## Abstract

Investigations that focused on the protective role of probiotics against Surgical Site Infections (SSI) in multiple-trauma (MT) patients are generally few, probably due to the complexity of the concept of trauma. We aimed to assess the efficacy of a four-probiotic regime to reduce the incidence of SSI in MT patients, with a brain injury included. MT patients, being intubated and expected to require mechanical ventilation for >10 days, were randomly allocated into placebo (*n* = 50) or probiotic treatment (*n* = 53) comprising *Lactobacillus acidophilus* LA-5 (1.75 × 10^9^ cfu), *Lactiplantibacillus plantarum* UBLP-40 (0.5 × 10^9^ cfu), *Bifidobacterium animalis subsp. lactis* BB-12 (1.75 × 10^9^ cfu), and *Saccharomyces*
*boulardii Unique-28* (1.5 × 10^9^ cfu) in sachets. All patients received two sachets of placebo or probiotics twice/day for 15 days and were followed-up for 30 days. The operations were classified as neurosurgical, thoracostomies, laparotomies, orthopedics, and others; then, the SSI and the isolated pathogen were registered. A total of 23 (46.0%) and 13 (24.5%) infectious insults in 89 (50 placebo patients) and 88 (53 probiotics-treated) operations (*p* = 0.022) were recorded, the majority of them relating to osteosynthesis—17 and 8, respectively. The most commonly identified pathogens were *Staphylococcus aureus* and *Acinetobacter baumannii*. Our results support published evidence that the prophylactic administration of probiotics in MT patients exerts a positive effect on the incidence of SSI.

## 1. Introduction

Surgical site infections (SSI) are currently one of the most challenging health care issues worldwide, being one of the most frequent types of nosocomial infections following any surgical procedure [1,2,3]. Today, more than 5% of patients undergoing a surgical procedure develop an SSI [4], defined as an infection occurring up to 30 days after surgery and affecting either the incision or deep tissue at the operation site [5]. A number of clinical studies have directly correlated SSI with changes in the gut microbial diversity due to surgical stress insult, with the loss of “health-promoting” commensal microbes and overgrowth of pathogenic bacteria, even in elective surgery patients [6,7,8,9,10,11]; microbial dysbiosis is believed to generally increase the susceptibility to nosocomial infections, sepsis, and organ failure [12].

Moreover, the issue of critically ill patients has been well-studied with respect to the gut microbial diversity and richness; dramatic microbiome alterations were found to occur within hours [13] of even a noninfectious stressor, such as myocardial infarction, stroke, or trauma [12], providing the opportunity for the pathogens to increase in abundance and virulence. However, in such critically ill patients, the most prominent and most studied infectious complication remains that of the lungs and, predominantly, ventilator-associated pneumonia [9,14,15,16].

Probiotics are defined as “live microorganisms that, when administered in adequate amounts, confer a health benefit on the host” via the restoration of commensal “healthy microbes” [17]; more specifically, they intervene by means of multiple pathways, including the suppression of pathogenic microbes, release of antimicrobial factors, modulation of immune cell proliferation, and enhancement of the gut epithelial barrier [17,18,19]. A recent meta-analysis of 34 randomized controlled trials reporting on 1354 participants who received probiotic or symbiotic preparations and 1369 controls revealed that the perioperative administration of either probiotics or synbiotics significantly reduced the risk of infectious complications following abdominal surgery [6], including a reduction in ventilator-associated pneumonia, bacteremia, length of hospital stay, and antibiotic use in surgical patients [20,21].

Multiple trauma (MT) patients, and especially those also with a traumatic brain injury (TBI), are among the most vulnerable patients in trauma care. From a trauma cohort of 1277 patients, 1001 infection cases (multiple infections in a number of patients), mainly respiratory, were reported in a total of 580 (45.4%) patients, leading to an overall mortality of 14.7% [22]. In an earlier study by our research group, a symbiotic formula (Synbiotic 2000 Forte) containing a four-probiotic combination (10^11^ cfu each): *Pediococcus pentosaceus* 5–33:3, *Leuconostoc mesenteroides* 32–77:1, *Lactobacillus paracasei* spp. *paracasei* 19, and *Lactobacillus plantarum* 2362 was administered in mechanically ventilated MT patients, resulting in a significant reduction in the overall infections and sepsis, mainly from *Acinetobacter baumannii*, as well as in the number of days on mechanical ventilation and ICU stay [23,24]. Additionally, a significant reduction of the CRP, procalcitonin, and endotoxin levels was also prominent in synbiotics-treated patients, this finding related to the delay in the advent of bloodstream infections [25]. 

In the recent ProVAP randomized clinical trial conducted by our study group [26], it was described that the intake of a preparation of four probiotics decreased significantly the incidence of ventilator-associated pneumonia (VAP) compared to placebo comparators. The majority of participants of ProVAP had MT and were subject to surgical operations before inclusion in the trial. We aimed at this subgroup of participants to investigate the post hoc impact of probiotic intake on the incidence of SSI. 

## 2. Materials and Methods

### 2.1. Study Design

This post hoc analysis was based on the data relating to surgical site infections, which were prospectively collected for the ProVAP trial (Clinicaltrials.gov NCT03074552) [26]. Briefly, the ProVAP trial is a multi-center, randomized, double-blind, placebo-controlled trial performed on multi-trauma patients who met the following entry criteria: (i) adults only, (ii) recent trauma involving brain injury and at least one more organ system and involving urgent intubation either in the ambulance car or the emergency department upon arrival, and (iii) the likelihood of mechanical ventilation duration ≥ 10 days and life expectancy ≥ 15 days. Exclusions to the study were described in detail in the previous publication dealing only with ventilator-associated pneumonia [26]. 

Patients eligible for participation, after written informed consent provided by a legal representative, were allocated to receive either a probiotic formula or placebo treatment, according to a computer-based randomization program blinded to the study investigators and the physicians in charge. The probiotic regime LactoLevure^®^ (Uni-Pharma SA, Athens, Greece) consisted of *Lactobacillus acidophilus* LA-5 (1.75 × 10^9^ cfu), *Lactiplantibacillus plantarum UBLP-40* (0.5 × 10^9^ cfu), *Bifidobacterium animalis* subsp. *lactis* BB-12 (1.75 × 10^9^ cfu) and *Saccharomyces boulardii Unique-28* (1.5 × 10^9^ cfu), while the placebo, looking identical to that of the probiotics, contained a powdered glucose polymer. Each patient received two sachets, twice daily, for 15 days, one by a nasogastric or gastrostomy tube into the stomach and the other as slurry to the oropharynx, hoping to be spread during passive swallowing towards the trachea and esophagus, as occurs with pathogens from the sinuses. Only the study participants who underwent surgical operations were included in this post hoc analysis.

SSI was defined as any infection presenting within 30 days after surgery involving the skin and subcutaneous tissue of the incision (superficial incisional) and/or the deep soft tissue (fascia or muscle) and necessarily accompanied by at least one of the following: (a) purulent drainage superficial or deep (fascia or muscle) but not from within the organ or space component of the surgical site; (b) pain or tenderness, localized swelling or redness, heat or fever, or several of these symptoms and the incision having been opened deliberately or by spontaneous dehisces; (c) abscess within the wound, clinically or CT-detected; and (d) the isolation of one typical pathogen [27,28]. Organ space infections were recorded separately and defined as intraabdominal or pelvic infections detected by the symptoms/clinical examination or by CT scan intraoperatively in the case of reoperations. The definitions of the other infections have been mentioned previously [26]. 

### 2.2. Patients

All patients who underwent an emergency operation upon admission, either to repair trauma or to investigate tissue damage, were initially recorded. Operations were classified according to the anatomic area into: (a) neurosurgery, mainly for cerebral decompression; (b) thoracostomies for pneumo/hemothotax; (c) exploratory laparotomy for the liver and/or spleen damage/hemorrhage or of other viscera injury; (d) orthopedics, for closed or open fracture reduction, and external or internal osteosynthesis; and (e) others, including severe facial fractures and vascular damage, related to open fractures.

On the post-operative day on which the SSI was recognized and registered, the isolated pathogen was also recorded and correlated with the same or previous days hospital-acquired pathogens (if any) from cultures of the peripheral blood and/or the catheter tip.

The number of patients having been operated on under a hemorrhagic shock status, either due to solid viscera rupture or due to a major fracture, was recorded and, finally, those with a history of diabetes mellitus and smokers.

### 2.3. Statistical Analysis

All the data were stored initially in a Microsoft Excel spreadsheet, while the statistical analysis was conducted with the help of the Statistical Package for Social Science (SPSS), Inc. (v 25.0; Chicago, IL, USA) after them being extracted. The normality of the data’s distribution was assessed by the Shapiro–Wilk test. We described both groups of patients by their sociodemographic (sex and age) characteristics, comorbidities, and scores in different scoring systems for critically ill and trauma patients with absolute and relative frequencies. The results of continuous variables were presented as the means ± standard deviation when normality was assumed or as medians with their respective interquartile range (Q25–Q75) when the data was skewed. For categorical variables, counts and percentages are presented. The Independent Student’s *t*-test was applied in order to compare the means between two independent samples with normally distributed data, whereas the Mann–Whitney *U* Test was used to compare the differences between two independent groups when the dependent variable was either ordinal or continuous and not normally distributed. Furthermore, the chi-square test was conducted for comparing the nominal data, and the odds ratio (OR) was calculated. Moreover, a Cox regression analysis was performed, and the Hazard Ratio of the incidence of SSI was calculated. Finally, statistical significance was set at *p* < 0.05.

## 3. Results

The original intent-to-treat population analyzed 53 patients allocated to treatment with the placebo and 59 patients allocated to treatment with probiotics. In the present post hoc analysis dealing with SSI, only patients subjected to at least one emergency operation on the day of admission were included; thus, 3 and 6 more patients were additionally excluded from the above groups, thus leaving 50 in the placebo group and 53 in the probiotic group. The study flow diagram is presented in Appendix A. The demographic characteristics of the patients are listed in Table 1. 

The patients were then analyzed in relation to the type and number of operations already performed before enrolment in the study (Table 2), the majority being subjected to osteosynthesis, 35 and 25 operations followed by thoracostomies, 20 and 26 operations, and exploratory laparotomies, 19 and 16 operations in placebo-treated and probiotic-treated patients. 

Almost half of the patients, 22 out of 50 (44%) and 26 out of 53 (49%), underwent one operation, *p* = 0.61; 17 (34%) and 19 (35.8%); two operations, *p* = 0.84; and 11 (22%) and 8 (15.1%) three operations, *p* = 0.37; thus, 50 placebo-treated patients underwent 89 operations and 53 probiotic-treated patients in 88 operations. The combinations of the different types of operations are shown in Table 3.

Twenty-three placebo-treated (46.0%) and 13 probiotic-treated patients (24.5%) experienced an SSI, the difference being statistically significant (*p* = 0.022). One major concern is whether most of the protection provided by probiotics is for patients who were subject to just one intervention or to more than one. This is analyzed in Table 4, showing that most captured SSI were related to osteosynthesis; 10 out of 14 placebo-treated patients (71.4%) and 2 out of 8 probiotic-treated (25.0%) (*p* = 0.035), followed by facial fractures (5 out of 7 (71.4%) and 2 out of 4 patients (50.0%) (*p* = 0.477) in the placebo and probiotic groups, respectively), all in the one-operation sub-group. The OR for SSI under probiotics was 0.11 (95% CIs 0.03–0.42; *p* < 0.0001) for patients undergoing one operation, 1.67 (95% CIs 0.33–8.35; *p*: 0.695) for patients undergoing two operations, and 0.72 (95% CIs 0.11–4.62; *p*: 1.00) for patients undergoing three operations. The time, post-operatively, to the appearance of SSI is presented in Figure 1.

It is of interest to comment that most SSI were found related to osteosynthesis; 10 out of 14 placebo-treated patients (71.4%) and 2 out of 8 probiotic-treated (25.0%) (*p* = 0.035), followed by facial fractures (5 out of 7 (71.4%) and 2 out of 4 patients (50.0%) (*p* = 0.477) in the placebo and probiotic groups, respectively), all in a one-operation sub-group. Quite similar, but not so excessive, were the findings in the two-operation group (Table 4). The time, post-operatively, to the appearance of SSI is presented in Figure 1.

The pathogens isolated in the two groups are tabulated in Table 5. It is noteworthy that only five placebo-treated and one probiotic-treated patients were found to have the same pathogen identified in both trauma and peripheral blood and/or central venous catheter tip cultures (*p* = 0.277). Another 14 and 3 patients (*p* = 0.029) in the two groups, respectively, were also found to have positive blood cultures at the time of SSI recognition, but the pathogen identified was different.

Finally, four patients from the placebo and three from the probiotic group experienced life-threatening hemorrhage; this was due to an open-book pelvic fracture in two cases, liver damage plus hemothorax in one, and a dual-hemothorax plus femur fracture in one, all four in placebo patients. Of equal severity were those in the probiotic group: an open, complicated fracture of a lower extremity leading, finally, to amputation plus facial fracture in one patient, spleen rupture and complicated femur fracture in one, and spleen rupture and hemothorax in one.

## 4. Discussion

In the present post hoc analysis, the early administration—within less than 6 h of admission—of a four-probiotic regimen containing *L. acidophilus* LA-5 1.75 × 10^9^ cfu, *L. plantarum* 0.5 × 10^9^ cfu, *B. lactis* BB-12 1.75 × 10^9^ cfu, and *S. boulardii* 1.5 × 10^9^ cfu was found to significantly reduce the incidence of SSI in severely ill, mechanically ventilated MT patients (rate 22.7% against 36.0%, *p* = 0.05). In the main publication that emerged from our experimental protocols [26], based on acute multiple-trauma patients with TBI and at least one more organ system trauma, the incidence of VAP was also found to be significantly lower in the probiotic-treated group in relation to the control (rate 11.9% versus 28.3%, *p* = 0.034), along with a delay in the onset of VAP, a shorter duration of ICU stay, and of hospital stay. 

Although both studies contained data from the same research protocol, in no case should the present analysis be considered as a duplication of the first one. The first study (VAP) was exclusively focused on the incidence ventilator-associated pneumonia and its prevention by means of the early administration of probiotics and with no reference to other infections; the present post hoc analysis referred exclusively to surgical site infections following emergency operations (performed upon admission] either to repair any trauma or just to investigate the tissue damage. There is no correlation between VAP and SSI, because the bacteria identified in SSI are quite different from those in bronchoalveolar lavage (BAL). Additionally, SSI infections have occurred earlier than VAP; 20% of the SSI presented up to day 10, as opposed to 20% of VAP presented after day 10 [26]. Additionally, most of the 7 (11.9%) and 15 (28.3%) of probiotic-treated and placebo patients, respectively, are not the same individuals as the 13 (24.5%) and 23 (46%) suffering SSI.

In the present study, all included MT patients also suffering TBI were operated on upon admission: 22 (44%), 17 (34%), and 11 (22%) of the placebo group and 26 (49%), 19 (35.8%), and 8 (15.1%) patients of the probiotic group underwent one, two, or three emergency operations in a total of 89 and 88 operations, respectively. Fifty patients from 89 operations experienced 23 SSI (25.8%%), and 53 probiotic-treated patients from 88 operations developed 13 SSI (14.8%), *p* = 0.067. The number of infections, although significantly lower in probiotic-treated patients, sounds high for both groups, given that the patients were mostly young, healthy, and without comorbidities, but it clearly reflects the type of surgeries: most patients presenting with at least one open fracture requiring internal or external fixation, and we should not underestimate the significance of the fact that all patients suffered a brain trauma, the severity of which required emergency intubation and mechanical ventilation. Furthermore, a brain injury of any kind promotes a decrease in *Bacteroidetes, Firmicutes*, and *Verrucomicrobia* and an increase in the *Clostridia* and *Enterococcus* populations [29,30,31]. Alterations in the gut microbiome composition favoring pathogenic over commensal bacteria have deleterious effects on both the CNS and the GI tract. The brain–gut axis is a bidirectional pathway that is critical for both the central nervous system and gut homeostasis, regulating diverse functions, including visceral pain, intestinal barrier function, gut motility, and neurobehavior [32]. When a TBI has occurred, the stress response is well-documented to negatively affect the autonomic nervous system impact on the control of GI functions [33].

Regarding fractures, the rate of infection after open reduction and internal fixation is generally recognized as 1–3% overall [34], whereas higher energy fractures and fractures of at-risk regions, including the tibia plateau, pilon, and calcaneus, can have rates of infection up to 50% in some situations [35,36], methicillin-resistant Staphylococcus aureus-positive nasal swab testing being one of the risk factors [37]. Although, in our protocols, such a test was not included, 6 out of 35 osteosyntheses in the placebo-treated and 9 out of 25 osteosyntheses in the probiotics-treated patients experienced a *Staphylococcus aureus* infection (*p* = 0.096). 

Much more complicated remains the situation for patients also with a huge, even life-threatening, hemorrhage, needing thus to receive inotropes and massive blood transfusion, and it is well-known that both blood cells themselves and the iron content promote pathogen growth [38]. In the present study, fortunately, only 4 and 3 patients from the control and probiotic groups, respectively, experienced a hemorrhagic shock upon admission, while almost all 50 and 53 patients received more than 3 units of whole blood in the very first few days after admission. 

Probiotics, although consumed for centuries by healthy individuals for GI health, are now gaining wide attention for use in hospital settings for the prevention of infectious complications. In recent years, a considerable number of studies have shown that the perioperative administration of probiotics and/or synbiotics, as a strategy to reduce dysbiosis, significantly diminishes the risk of infectious complications following surgery, with the magnitude of this risk reduction approaching 50% [6,7,20,21,39]. A recent meta-analysis of 35 trials and 3028 patients was the first one to exclusively investigate the effect and possible mechanism of action of pro-/synbiotics to lower the risk of SSIs [39]. According to a recent literature review, Lukic et al. [40] described three possible mechanisms by which oral probiotic treatment accelerates wound healing: (i) through immunomodulation: intestinal probiotics stimulate the recruitment of lymphocytes to the injured tissue, contributing to the activation of innate and adaptive immune responses [41,42]; furthermore, they can positively influence skin health by increasing the number of γδ T cells and Th17 cells in the spleen or axillary lymph nodes and, thus, the production of proinflammatory cytokines by T cells, which, in turn, accelerate the anti-inflammatory cutaneous effects [40]. Especially, *L. plantarum* and *L. fermentum* were shown to induce the phagocytic activity of PMNs in the peripheral circulation, which is related to the induction of granulocyte–macrophage colony-stimulating factor (GM-CSF) production [43]; (ii) through the improved absorption of essential nutrients: vitamins, minerals, and enzyme cofactors are involved in tissue repair to heal skin wounds; *L. reuteri* and *L. acidophilus* were shown to increase the absorption of dietary vitamins D and E, known to be important for wound healing [40]; and (iii) through the central nervous system: probiotics produce neuroactive molecules and/or modulate the secretory activity of intestinal mucosal enteroendocrine cells, leading to the release of neuromodulators with the potential to improve tissue regeneration [40]. 

However, it is well-known that the different host benefits and, especially, the immunomodulatory effects accelerated by probiotics are species-, strain-, dose-, and probably time-specific [44,45]. It has been shown that *L. plantarum*—one of the ingredients of the probiotic regimen used in the present study—increased the phagocytic activity of peritoneal macrophages [46], while others stimulate macrophage activity [40] and induce the expression of different growth factors (TGF-β, VEGF, EGF, EGFR, and IGF] [47,48]. Generally speaking, *Lactobacillus* strains are capable of inducing proinflammatory cytokines such as IL-12 and IFN-γ in addition to anti-inflammatory cytokines such as IL-10 [49], whereas *Bifidobacterium* strains are generally better inducers of IL-10 than *Lactobacillus* strains [44,50]. This is why the combination of species is preferable, as in our study, where the well-tested, in previous RCTs of our study group [26,51,52], LactoLevure^®^ regime was used.

Besides the anti-inflammatory and immune-modulatory effects, another benefit of probiotics is the improvement of tissue repair by means of positive stimulation of the wound healing process. Although this process has not yet been completely investigated in relation to surgical trauma, this is of high priority, since speedy trauma repair is vital to prevent the entrance of harmful microorganisms to avoid blood loss and body dehydration and mainly to promptly restore skin barrier function. In a recent experimental study of our group [53], the topical application of *Lactobacillus plantarum* to excisional wounds was found to start the healing process much earlier than the combined treatment with *L. rhamnosus* plus *Bifidobacterium longum*, the wounded area, on day 4, being reduced from 41.2% to 29.5% (*p* = 0.0011), while Poutahidis et al. [54] reported that, in *Lactobacillus reuteri*-treated mice, the cutaneous wound healing process was accelerated two-fold due to the upregulation of the neuropeptide hormone oxytocin by a vagus nerve-mediated pathway.

The study protocol did not include a skin microbiome analysis or surgical wound closure inspection. However, there were significantly fewer surgical infections in the probiotic-treated patients, which is hard to attribute to other than the improvement of the total body microbiome diversity. It is also possible that probiotic administration also leads to earlier trauma re-epithelialization and collagen deposition. The increased immune response of the patent/host results in faster wound healing as a consequence of the trauma remaining uninfected or as a positive effect of the probiotics on the intestinal microbiome, leading to accelerated wound healing.

Finally, it is noteworthy that no patient in either group experienced an infection related to probiotics treatment, as Hempel et al. [55] also concluded after an exhaustive literature review. Furthermore, although the total in-hospital days of stay is not directly affected by the surgical trauma infection, there was a significant reduction in the probiotics group, this difference perhaps being more globally related to the well-being of an individual receiving probiotics.

Our study had some limitations: (i) it was a post hoc analysis; after we published the results on VAP prevention, we realized that there was a gap in the literature regarding the influence of probiotics on surgical site infections after emergency operations in multiple traumatized patients; (ii) the study was powered as the primary outcome being the reduction of the incidence of ventilator-associated pneumonia; and (iii) there is a statistically significant difference in the number of osteosyntheses performed, either as an external or internal fixation, as well as the number of fractures per patient.

## 5. Conclusions

The results of the present study support published evidence that the prophylactic administration of probiotics exerts a positive effect on the incidence of surgical site infections in severe multiple trauma—involving injury of the brain and at least one more organ system—patients being urgently intubated and under ventilatory support. The four-probiotics regime used continues to be a safe measure to fight microbial invasions in surgical traumas in patients experiencing microbial dysbiosis after stressful stimuli, as is an acute multiple trauma plus surgical stress for restoration. The evidence indicates that probiotics should hold a strong position in the treatment regime for the management of critically traumatized patients.

## Figures and Tables

**Figure 1 nutrients-14-02620-f001:**
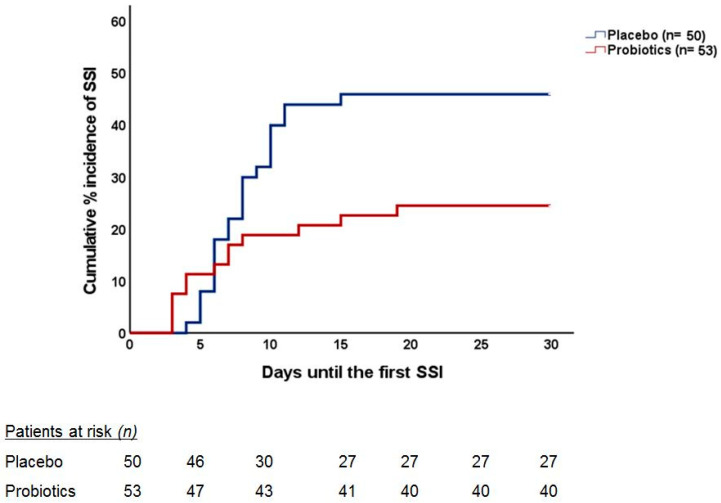
Cumulative incidence of SSI occurrence during the postoperative period in each group. The curves indicate the time to the incidence of the first episode of SSI in each group. The number of patients at risk over follow-up is indicated. The two curves are compared using Cox regression analysis; the Hazard Ratio (HR) of the incidence of SSI under probiotic intake compared to placebo intake is provided alongside the respective confidence intervals (CIs); the *p*-value of the difference between the two curves is also provided.

**Table 1 nutrients-14-02620-t001:** Demographic data.

	Placebo (*n* = 50)	Probiotics (*n* = 53)	*p*-Value
Male gender, n (%)	40 (80.0%)	50 (94.3%)	0.028
Age (years), mean ± SD	44.1 ± 13.9	38.4 ± 16.9	0.061
Smokers, n (%)	13 (26.0%)	12 (22.6%)	0.691
Diabetes Mellitus, n (%)	4 (8.0%)	5 (9.4%)	0.796
Total number of operations	89	88	-
* NISS, mean ± SD	7.68 ± 2.17	7.43 ± 2.13	0.443
* GCS, mean ± SD	10.02 ± 4.17	10.81 ± 3.49	0.133
* SOFA score, mean ± SD	6.22 ± 1.41	5.91 ± 1.44	0.142
* APACHE II score, mean ± SD	15.28 ± 5.61	14.82 ± 5.22	0.496

SD: Standard Deviation, NISS: New Injury Severity Score, GCS: Glasgow Coma Scale, SOFA: Sequential Organ Failure Assessment, and APACHE: Acute Physiology and Chronic Health Evaluation. * All the scores presented in the table have been calculated at the time of patients’ admission in the intensive care unit. “-”: *p* value cannot be calculated.

**Table 2 nutrients-14-02620-t002:** Operations, classified according to the anatomic area per treatment group.

Type of Operation	Placebo (*n* = 50)	Probiotics (*n* = 53)	*p*-Value
exploratory laparotomy	19	16	0.40
neurosurgery	6	13	0.10
osteosynthesis	35	25	0.019
thoracostomies	20	26	0.36
others	9	8	0.69
**Total number of operations**	**89**	**88**	

**Table 3 nutrients-14-02620-t003:** The combination of operations performed per treatment group.

Type of Operations Per Patient	Placebo	Probiotics	*p*-Value
**ONE OPERATION/PATIENT**			
Osteosynthesis	14	8	0.11
Neurosurgery	1	7	0.03
Laparotomy	0	4	0.05
Thoracostomy	0	3	0.08
Others	7	4	0.29
**Number of patients, n (%)**	**22 (44.0%)**	**26 (49.0%)**	0.61
**Number of operations, n**	**22**	**26**	0.47

**TWO OPERATIONS/PATIENT**			
Osteosynthesis + thoracostomy	4	11	0.07
Osteosynthesis + laparotomy	7	1	0.02
Osteosynthesis + others	1	1	0.97
Laparotomy + neurosurgery	0	1	0.33
Laparotomy + thoracostomy	5	4	0.66
Laparotomy + others	0	1	0.33
**Number of patients, n (%)**	**17 (34.0%)**	**19 (35.8%)**	0.84
**Number of operations, n**	**34**	**38**	0.50

**THREE OPERATIONS/PATIENT**			
Neurosurgery + thoracostomy + osteosynthesis	4	2	0.36
Neurosurgery + thoracostomy + laparotomy	1	3	0.34
Thoracostomy + laparotomy + osteosynthesis	5	1	0.08
Thoracostomy + laparotomy + others	1	1	0.97
Thoracostomy + osteosynthesis + others	0	1	0.33
**Number of patients, n (%)**	**11 (22.0%)**	**8 (15.2%)**	0.37
**Number of operations, n**	**33**	**24**	0.16

**Total number of patients, n (%)**	**50**	**53**	-
**Total number of operations, n**	**89**	**88**	-

**Table 4 nutrients-14-02620-t004:** SSI infections occurred in relation to the number and type of operations in each group.

	PLACEBO		PROBIOTICS		
OPERATIONS	Patients	SSI	Patients	SSI	*p*-Value
**ONE OPERATION/PATIENT**					
Osteosynthesis	14	10	8	2	0.035
Neurosurgery	1	0	7	0	n/a
Laparotomy	0	0	4	1	n/a
Thoracostomy	0	0	3	0	n/a
Others	7	5	4	2	0.477
**Number of patients**	**22**		**26**		0.61
**Number of SSI**		**15**		**5**	<0.001

**TWO OPERATIONS/PATIENT**					
Osteosynthesis + thoracostomy	4	2	11	2	0.22
Osteosynthesis + laparotomy	7	0	1	2	n/a
Osteosynthesis + others	1	0	1	1	0.157
Laparotomy + neurosurgery	0	0	1	0	n/a
Laparotomy + thoracostomy	5	1	4	0	0.12
Laparotomy + others	0	0	1	0	n/a
**Number of patients**	**17**		**19**		0.84
**Number of SSI**		**3**		**5**	0.53

**THREE OPERATIONS/PATIENT**					
Neurosurgery + thoracostomy + osteosynthesis	4	2	2	1	0.99
Neurosurgery + thoracostomy + laparotomy	1	0	3	0	n/a
Thoracostomy + laparotomy + osteosynthesis	5	3	1	1	0.43
Thoracostomy + laparotomy + others	1	0	1	1	n/a
Thoracostomy + osteosynthesis + others	0	0	1	0	n/a
**Number of patients**	**11**		**8**		0.37
**Number of SSI**		**5**		**3**	0.73

**Total number of patients**	**50**		**53**		
**Total number of SSI**		**23**		**13**	0.022

n/a: Not applicable (chi-square test cannot be calculated).

**Table 5 nutrients-14-02620-t005:** Number of pathogens isolated from the surgical traumas.

PATHOGENS n (%)	Placebo	Probiotics	*p*-Value
*Staphylococcus aureus*	6 (25.0)	9 (69.2)	0.009
*Proteus mirabilis*	4 (16.7) *	-	0.110
*Acinetobacter baumannii*	5 (20.8) *	-	0.078
*Enterococcus faecium*	-	1 (7.7)	0.168
*Pseudomonas aeruginosa*	4 (16.7)	1 (7.7)	0.446
*Klebsiella oxytoca*	3 (12.5)	1 (7.7)	0.653
*Serratia marcescens*	2 (8.3)	1 (7.7)	0.946
**Total *n* of pathogens isolated**	**24**	**13**	

* One placebo-treated patient had two pathogens isolated (Proteus and Acinetobacter).

## Data Availability

The data and materials/figures used in the current study are available from the corresponding author upon reasonable request.

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
