# Peer review of "A Four-Probiotic Regime to Reduce Surgical Site Infections in Multi-Trauma Patients"

_nutrients, 2022, doi:10.3390/nu14132620_

Round 1

Reviewer 1 Report

In this study, the authors examined the effect of probiotic administration on the development of SSI in multi-trauma patients who received placebo or probiotics and among whom surgery was performed. Since this is a post-hoc study, it is difficult to confirm the effect of probiotic intake, but I think it is very important for the accumulation of clinical cases and is a very challenging study.

On the other hand, the authors, in reporting a study dealing with probiotics, do not discuss it based on the assumption that the effects of probiotics are normally strain-specific. Although this article is for the Special Issue, I think that the superiority and significance of this research report will not be conveyed to the readers of 'Nutrients'. Therefore, I request to amend the manuscript.

[Major comments]

1. The authors should state the name of probiotics strains used. For example, Lactobacillus plantarum (currently Lactiplantibacillus plantarum) is the scientific name indicating species, and any Lactobacillus plantarum that the authors actually administered must be strains. It is scientifically important to state the strain name. Also, the effects of probiotics are essentially strain-specific. In the Discussion, the authors discuss the various effects of probiotics as if they were at the species level, but most of the cited studies probably only showed effects at the strain level. Therefore, for those that are not at the species level, the strain name should be described. Then, please discuss why you used the four strains in this study and compare their superiority to other strains.

2. The authors state in the Discussion (Line 224-230) as below;

‘Although both studies contain data from the same research protocol, in no case the present analysis could be considered as a duplication of the first one. The first study [VAP] was exclusively focused on the incidence ventilator associated pneumonia and its prevention by means of early administration of probiotics and with no reference to other infections; the present posthoc analysis is referred exclusively to surgical site infections following emergency operations [performed upon admission] either for to repair any trauma or just to investigate tissue damage, and no correlation has been made to VAP.’

    The authors should provide data to show that there is no correlation with VAP. Also, as this study at least examined along with the ProVAP protocol, data of VAP should be added. Furthermore, the Study Flow Chart should be added to the figure or included as a supplementary material to clarify the difference from the previously reported studies (26).

3. In the Discussion, the parts (Line 232-255) seem to be mostly what should be described in the Introduction. If the authors would like to include them in the Discussion, please discuss properly how they relate to the data obtained in this study.

[Minor comments]

1.  Italicize bacterial names in the Abstract.

2. The definition of ‘probiotics’ in the reference 17 has been modified by ISAPP. The revised definition (which=>that) should be used. Nat Rev Gastroenterol Hepatol. 2014, 11(8):506-14.

Author Response

Response to Reviewer 1

Point 1: The authors should state the name of probiotics strains used. For example, Lactobacillus plantarum [currently Lactiplantibacillus plantarum] is the scientific name indicating species, and any Lactobacillus plantarum that the authors actually administered must be strains. It is scientifically important to state the strain name. Also, the effects of probiotics are essentially strain-specific. In the Discussion, the authors discuss the various effects of probiotics as if they were at the species level, but most of the cited studies probably only showed effects at the strain level. Therefore, for those that are not at the species level, the strain name should be described. Then, please discuss why you used the four strains in this study and compare their superiority to other strains.

Response 1: We thank the reviewer for this insightful comment. We totally agree with the reviewer’s opinion. Thus, we stated the strain name of the probiotics administered to the intervention group [changes are shown in lines 22-23 [abstract] and lines 98-101].

Regarding the use of the four strain probiotic regime, we decided to administer it because: it is a commercially available formulation in our country; the manufacturer kindly provide us with the placebo form, in an identical packaging; and we have sufficient experience of its beneficial effects in RCT’s performed by our group in colon cancer patients-colectomies [World Journal Surg 2015;39:2776], in Helicobacter pylori erradication [Nutrients 2022;14: 632] and recently in ventilator-associated pneumonia [Intern J Antimicrob Agents 2022;59: 106471]. We discuss it in “Discussion”, line 325 and thereafter.

Point 2: The authors state in the Discussion [Line 224-230] as below; ‘Although both studies contain data from the same research protocol, in no case the present analysis could be considered as a duplication of the first one. The first study [VAP] was exclusively focused on the incidence ventilator associated pneumonia and its prevention by means of early administration of probiotics and with no reference to other infections; the present posthoc analysis is referred exclusively to surgical site infections following emergency operations [performed upon admission] either for to repair any trauma or just to investigate tissue damage, and no correlation has been made to VAP.’

    The authors should provide data to show that there is no correlation with VAP. Also, as this study at least examined along with the ProVAP protocol, data of VAP should be added. Furthermore, the Study Flow Chart should be added to the figure or included as a supplementary material to clarify the difference from the previously reported studies [26].

Response 2: We appreciate the reviewer for this comment. There is no correlation between VAP and SSI because, bacteria identified in SSI are quite different from those in BAL. Additionally, SSI infections have occurred earlier than VAP; the 20% of SSI’s presented up to day 10, as oppose to 20% of VAP’s presented after day 10 [Intern J Antimicrob Agents 2022;59: 106471] [please see lines 259-264]. Moreover, most of the 7 [11.9%] and 15 [28.3%] of probiotic-treated and placebo patients, respectively, are not the same individuals, as the 13 [24.5%] and 23 [46%] suffering SSI. Furthermore, data from the ProVap protocol are completely available in the MedLine, and therefore we did not include any of them, considering that this would be quite redundant or even duplication. We added the flow chart for this post-hoc analysis [Supplementay Material: Figure S1, line 160], as the reviewer suggested.

Point 3: In the Discussion, the parts [Line 232-255] seem to be mostly what should be described in the Introduction. If the authors would like to include them in the Discussion, please discuss properly how they relate to the data obtained in this study.

Response 3: We have omitted the lines 232-255 (which are represented in lines 265-289 in the revised manuscript) according to Reviewer’s suggestions

Point 4: Italicize bacterial names in the Abstract

Response 4: We have made these changes [lines 22-24 and 9-30]

Point 5: The definition of ‘probiotics’ in the reference 17 has been modified by ISAPP. The revised definition [which=>that] should be used. Nat Rev Gastroenterol Hepatol. 2014, 11[8]:506-14.

Response 5: We have changed it accordingly [lines 52-54] and we have added the proper reference [Reference number 17].

Reviewer 2 Report

Introduction:

Line 79: Should Mt be MT?

Methods:

Lines 100-102: Please further explain exactly how the probiotics were administered and where they were thought to end up? Gut? Ip cavity? Sinuses? In other words, why were the probiotic sachets administered where they were? As opposed to directly on the skin where a potential SSI might have occurred.

Results:

Line 174: p = 0.022 is not “highly” significant. Would suggest removing “highly”.

Line 181:  typo – “…patients undergoing on operation;1.67…” on should be one?

Line 191: Table 4 – bottom of table should list an additional statistic: Total number of patients with SSI and statistical test should be run on those proportions; rather than only against total number of patients. This would eliminate erroneous counting of patients that had no SSI in the statistical analysis comparing placebo vs. probiotic. This additional line in the table would be more compelling and is it still statistically significant? In other words: Placebo – total number of infections (23) / total number of patients with SSI (39) = 59%. Probiotics – total number of infections (13) / total number of patients with SSI (33) = 39%.

Line 195: Figure 1 caption needs more information explaining the details of the figure. While interesting, it’s difficult to ascertain the full significance of the finding without more detailed information regarding the data depicted. For example, the n = in the figure legend is the number of patients? And the n = across days in the figure is the # of SSI’s? Please clarify. What is the p = 0.043 referring to? One would assume the reduction of % incidence of SSI, but it’s not clear.

Line 202: Even though chi-squared can’t be calculated on some individual pathogens; it can be calculated on the Total n of pathogens isolated, correct? With acknowledgement that not all pathogens were found in both groups.

Discussion:

Line 224: missing word? “…in no case (should) the present….”

Lines 340-342: Authors state that reduction of surgical infections could undeniably be attributed to improvement of total body microbiome diversity. This is a bit of a strong conclusion given that the “total body microbiome diversity” was not measured. This can actually be measured, but was not in this study, thus authors should rephrase this sentence.

Lines 342-344: Authors again state as per above, but this wasn’t measured in this study? Was this measured in a prior study? Also, the probiotic preparation…increases the immune response of patient/host….; this wasn’t measured in this study either as written here? Please clarify if these sorts of measurements were done in a prior publication, or rephrase.

Author Response

Response to Reviewer 2

Point 1: Introduction: Line 79: Should Mt be MT?

Response 1: We have changed it accordingly [line 79].

Point 2: Methods: Lines 100-102: Please further explain exactly how the probiotics were administered and where they were thought to end up? Gut? Ip cavity? Sinuses? In other words, why were the probiotic sachets administered where they were? As opposed to directly on the skin where a potential SSI might have occurred.

Response 2: Probiotic was administered: [i] by nasogastric or gastrostomy tube towards the stomach, and [ii] as slurry to the oropharynx, hoping to be spreaded during passive swallowing towards trachea and esophagus, as occurs with pathogens from the sinuses [lines 104-106]. To use probiotics directly on the skin is an attractive option – we have tested it in excisional skin wounds, in rats [Injury. 2022;53[4]:1385]. But, in the present situation [i] there is no experimental or clinical data for prevention of tissue infections after topical application; and [ii] technically, it is difficult to apply probiotic “cream” on healthy skin suture line. In any case our protocol [ProVap] was initially scheduled for probiotics to protect from VAP when administered through the gut and through the upper aerodigestive.

Point 3: Results: Line 174: p = 0.022 is not “highly” significant. Would suggest removing “highly”.

Response 3: We have removed “highly” [line 191].

Point 4: Line 181:  typo – “…patients undergoing on operation;1.67…” on should be one?

Response 4: We thank the reviewer for his/her meticulous evaluation of our manuscript. We have changed “on” to “one” [line 198].

Point 5: Line 191: Table 4 – bottom of table should list an additional statistic: Total number of patients with SSI and statistical test should be run on those proportions; rather than only against total number of patients. This would eliminate erroneous counting of patients that had no SSI in the statistical analysis comparing placebo vs. probiotic. This additional line in the table would be more compelling and is it still statistically significant? In other words: Placebo – total number of infections [23] / total number of patients with SSI [39] = 59%. Probiotics – total number of infections [13] / total number of patients with SSI [33] = 39%.

Response 5: Infections and SSI is exactly the same, thus we really have 23 infections [SSI] in 50 probiotic-treated patients and 13 SSI [infections] in 53 placebo-treated: p being 0.022. In Table 4 we have changed the word “infection” with the SSI, to avoid mis-understanding.

Point 6: Line 195: Figure 1 caption needs more information explaining the details of the figure. While interesting, it’s difficult to ascertain the full significance of the finding without more detailed information regarding the data depicted. For example, the n = in the figure legend is the number of patients? And the n = across days in the figure is the # of SSI’s? Please clarify. What is the p = 0.043 referring to? One would assume the reduction of % incidence of SSI, but it’s not clear.

Response 6: The reviewer is right. We feel sorry about this mis-understanding. A revised Figure 1, with a fully detailed legend to it, are now provided (lines 215-219).

Point 7: Line 202: Even though chi-squared can’t be calculated on some individual pathogens; it can be calculated on the Total n of pathogens isolated, correct? With acknowledgement that not all pathogens were found in both groups.

Response 7: The reviewer’s aspect is also correct and probably more suitable for the presentation of our results. Thus, we have compared again the data of the isolated pathogens [using chi-square test] calculated on the total number of the pathogens isolated; and we have changed, accordingly, the p-values in Table 5 [page 7]. Based on this, all the chi-squared was calculated for all the pathogens.

Point 8: Discussion: Line 224: missing word? “…in no case [should] the present….”

Response 8: No missing word – we changed “could” with “should”.

Point 9: Lines 340-342: Authors state that reduction of surgical infections could undeniably be attributed to improvement of total body microbiome diversity. This is a bit of a strong conclusion given that the “total body microbiome diversity” was not measured. This can actually be measured, but was not in this study, thus authors should rephrase this sentence.

Lines 342-344: Authors again state as per above, but this wasn’t measured in this study? Was this measured in a prior study? Also, the probiotic preparation…increases the immune response of patient/host….; this wasn’t measured in this study either as written here? Please clarify if these sorts of measurements were done in a prior publication, or rephrase.

Response 9: We aggree with the Reviewer’s opinion. Thus, we have changed this paragraph (lines 338-347) as follows (lines 374-381):

“The study protocol did not include skin microbiome analysis or surgical wound closure inspection. However, there were significantly fewer surgical infections in the probiotic-treated patients, which is hard to attribute to other than the improvement of the total body microbiome diversity. It is also possible that probiotic administration also leads to earlier trauma re-epithelialization and collagen deposition. The increased immune response of the patent/host results in faster wound healing, as a consequence of the trauma remaining uninfected or as a positive effect of the probiotics on the intestinal microbiome, leading to accelerated wound healing.”

Round 2

Reviewer 1 Report

The authors responded sincerely to my comments, and I understood their points.